# An Evaluation of the Sensitivity and Applicability of a Droplet Digital Polymerase Chain Reaction Assay to Simultaneously Detect *Pseudomonas aeruginosa* and *Pseudomonas fragi* in Foods

**DOI:** 10.3390/foods13101453

**Published:** 2024-05-08

**Authors:** Ju Huang, Ligong Zhai, Junyin Wang, Xiaotian Sun, Baoshi Wang, Zhaohui Wei

**Affiliations:** Department of Food Engineering College, Anhui Science and Technology University, Chuzhou 233100, China; hj2021777@163.com (J.H.); wangjy@ahstu.edu.cn (J.W.); 15398173254@163.com (X.S.); wangbaoshifsd@126.com (B.W.); weizhaohui109@163.com (Z.W.)

**Keywords:** *Pseudomonas aeruginosa*, *Pseudomonas fragi*, simultaneous detection, droplet digital PCR

## Abstract

Achieving effective control over microbial contamination necessitates the precise and concurrent identification of numerous pathogens. As a common bacterium in the environment, *Pseudomonas* is rich in variety. It not only has pathogenic strains, but also spoilage bacteria that cause food spoilage. In this research, we devised a remarkably sensitive duplex droplet digital PCR (dddPCR) reaction system to simultaneously detect pathogenic *Pseudomonas aeruginosa* (*P. aeruginosa*) and spoilage *Pseudomonas fragi* (*P. fragi*). By employing comparative genomics, we identified four genes of P. fragi. Through a specific analysis, the *RS22680* gene was selected as the detection target for *P. fragi*, and the *lasR* gene was chosen for *P. aeruginosa*, which were applied to construct a dddPCR reaction. In terms of specificity, sensitivity and anti-interference ability, the constructed dddPCR detection system was verified and analyzed. The assay showed excellent sensitivity and applicability, as evidenced by a limit of detection of 10^0^ cfu/mL. When the concentration of natural background bacteria in milk or fresh meat was 100 times that of the target detection bacteria, the method was still capable of completing the absolute quantification. In the simulation of actual sample contamination, *P. aeruginosa* could be detected after 3 h of enrichment culture, and *P. fragi* could be detected after 6 h. The established dddPCR detection system exhibits exceptional performance, serving as a foundation for the simultaneous detection of various pathogenic bacteria in food products.

## 1. Introduction

As a common food safety hazard, microbial contamination seriously affects human life and health [1]. There are many kinds of microorganisms with different sources, and the same bacteria can include many taxonomically related species. The *Pseudomonas* complex group has been called a “hodgepodge” for decades; it contains *P. aeruginosa*, *P. fragi*, *Pseudomonas fluorescens*, etc. [2,3]. *Pseudomonas* is a kind of aerobic bacteria; some of them are conditional pathogenic bacteria, and some are spoilage bacteria. Conditional pathogenic bacteria can form biofilm that will enhance resistance to traditional bactericidal methods, such as ultraviolet, drying and commonly used chemical disinfectants [4]. Spoilage bacteria have strong protein and lipid hydrolysis ability, making it easy to reduce meat quality and cause resource waste [5]. Therefore, it is necessary to identify different genera of *Pseudomonas*, which is similar to the identification of different genera of *Listeria*, in order to distinguish their pathogenicity [6]. It is well known that *P. aeruginosa* is a common bacterium that is widely found in water and other environments in nature [7]. It has strong drug resistance and high temperature resistance and prefers humid environments [8]. Therefore, it is easy to cause *P. aeruginosa* contamination during food processing and storage. Studies have shown that the detection rate of *P. aeruginosa* in drinking water in China can reach 10% [9]. Legesse Garedew et al. isolated and identified 54 kinds of bacteria in milk containers, of which 18.5 % were *P. aeruginosa* [10]. In addition, due to drug resistance and dense biofilms, *P. aeruginosa* has a very high growth advantage in animal-derived foods [11,12,13]. Although it does not have a direct fatal hazard to the human body, it may cause vomiting, diarrhea, fever and other symptoms after infection. Therefore, the pollution of *P. aeruginosa* must be strictly controlled.

In contrast, *P. fragi* is known as a “specific spoilage organism” which is abundant in chilled meat [14]. It can survive for a long time at a low temperature and decompose the protein in food, which makes the food lose its original freshness and taste, seriously affecting the appetite of consumers [15,16]. It is reported that 21% of the huge losses of meat products are caused by microbial spoilage [17]. When microorganisms work together to contaminate food, there is a situation in which some strains dominate. Zhang et al. found that *P. fragi* exhibited a clear predominance in cold chain food [9]. Wang et al. discovered that *P. fragi* showed the strongest spoilage potential in chilled chicken [18]. In addition, due to its rich biofilm, *P. fragi* can easily contaminate aquatic products such as salmon [19,20]. It is likely to cause contamination during food processing, transportation and storage, especially for fresh, refrigerated and frozen foods that have not been subjected to high temperature treatments or other forms of disinfection or preservation.

At present, the gold standard for the identification of pathogens is bacteriological culture, which is complex, time-consuming and unable to detect strains that are difficult to culture or lack specificity [21]. More simple and accurate detection methods, such as molecular detection, enzyme-linked immuno-sorbent assay and electrochemical detection, have been developed [22,23]. Molecular methods have shown great excellence in accurate detection [24]. Comparative genomics methods are used to detect bacteria based on specific nucleic acid sequences. However, the accuracy of these methods for *P. aeruginosa* is not sufficient. A large number of specific sequences were used to verify the specificity of molecular experiments to detect *P. aeruginosa* [25]. Furthermore, Murugan et al. used multiple pairs of primers to detect *P. aeruginosa* by mPCR in order to determine the accuracy [26]. So far, the research on *P. fragi* in the literature has focused on its genes and mechanisms [27]. Therefore, it is necessary to develop more sensitive and accurate molecular detection methods for the identification of microorganisms.

Digital PCR technology is continuously improved on the basis of polymerase chain reaction [28]. It can achieve absolute quantitative detection and shows great superiority in molecular detection. Digital PCR divided a reaction system into a large number of independent micro-reaction units, and the nucleic acid copy number was calculated according to the Poisson distribution and the positive ratio [29]. This method can accurately detect the target bacteria and has a significant advantage in accurately judging complex flora [30]. *P. aeruginosa* is a pathogenic bacterium, and *P. fragi* is a spoilage bacterium. Both of them belong to the *Pseudomonas* Genus and can endanger food. To satisfy the need for accurate, sensitive and multiplex detection, this study established a molecular method for testing *P. aeruginosa* and *P. fragi* at the same time in the same device. Using the method of comparative genomics, *RS22680* and *lasR* were identified as detection targets. Primers and probes were designed based on these two genes, and dddPCR detection was constructed. The effectiveness of the reaction system was proven by sensitivity, anti-interference ability experiments and artificially simulated contaminated samples. This method can help to identify the types and quantities of bacteria in food, achieving food quality control and reducing food loss.

## 2. Material and Methods

### 2.1. Sample Preparation

Samples (raw chicken and potable water) were purchased from a fresh supermarket in Fuxi Street, Fengyang County, Chuzhou City, Anhui Province, China. The raw chicken was cut into 25 g and frozen at −20 °C for subsequent experiments. Fresh milk was obtained from Dutch dairy cows (Heping Dairy Ranch, No. 1151, South Yan An Road, Bengbu City, Anhui Province, China) through aseptic sampling and sent back to the laboratory for processing as soon as possible at a low temperature. For the latter experiments of artificial pollution commercial sample, purchased drinking water, sterile milk and raw chicken were identified by microbial culture method without *P. aeruginosa* or *P. fragi* [31].

### 2.2. Strain Culture and DNA Extraction

The strains used in this experiment were all from standard strains purchased from formal channels. *P. aeruginosa* was cultured in a Luria–Bertani (LB) broth at 37 °C for 18 h, and *P. fragi* was cultured at 30 °C. Other bacteria used for specificity analysis were activated according to the culture instructions. DNA was extracted using the bacterial genome extraction kit (Shanghai Sangon Biotech, Shanghai, China), and the concentration was determined under a spectrophotometer (NV3000C, Vastech Inc., San Jose, CA, USA) and stored at −20 °C. Genomic DNA of chilled meat was extracted by modifying the direct lysis (DL) method [32]. The sample solution was mixed by ultrasonic treatment for 5 min and incubated for 10 min in a boiling water bath. Finally, the sample was centrifuged at 10,000 rpm for 5 min, and the supernatants were collected as the reaction template.

### 2.3. Screening of Specific Genes of P. aeruginosa and P. fragi

Three whole genome sequences of *P. fragi* (GeneBank: GCA_002128325.1, GCA_02986945.1 and GCA_000250595.1) were obtained from NCBI (https://www.ncbi.nlm.nih.gov/, accessed on 17 September 2022). Sequence alignment of *P. fragi* was performed using NCBI Nucleotide-BLAST (https://blast.ncbi.nlm.nih.gov/Blast.cgi, accessed on 17 September 2022). Each CDS of *P. fragi* was matched using BLASTN, and those exhibiting low homology with non-*Pseudomonas* spp. and high homology with all *P. fragi* (E-value < 1 × 10^−200^, Query Cover ≥ 99%) were used as candidate detection targets. *lasR*, *gyrB* and *rpoB* were finally selected as the *P. aeruginosa* candidate genes according to the reported literature [33,34]. The specificity of genes was analyzed using NCBI Primer-BLAST. The genetic information involved in this paper is shown in Table 1.

In order to ensure the accuracy of specific genes, primers were designed according to candidate genes, and 20 strains of non-*Pseudomonas fragi* and non-*Pseudomonas aeruginosa* were used for comparative analysis. The specificity result was analyzed by PCR experiments and agarose gel electrophoresis imaging. The primers used in the experiment are shown in Table 1, and the PCR reaction was performed in 25 μL amplification mixture containing 1 μL of the DNA templates, 12.5 μL 2 × Reaction Mix (Dongsheng Biotechnology Co., Ltd. Guangzhou, China), 1 μL each of primers F and R (10 μM) and 8.5 μL sterilized ultrapure water.

### 2.4. Primers, Probe Design for dddPCR and Specificity Verification

The specific genes of *P. aeruginosa* and *P. fragi* were screened, and primers and probes were designed according to the experimental requirements based on the highly conserved region, as shown in Table 2. The primers and probes were designed by primer 3.0 and synthesized and purified in Sangon Biotech, Shanghai, China. In order to identify the accuracy of the primers designed in the experiment, the specificity of digital PCR primers was verified by qPCR for common bacteria and other *Pseudomonas*. This includes 5 other *Pseudomonas* strains and 5 common bacteria. The genomes of these 10 strains were used as templates for reaction, and the results of the qPCR were used to determine the specificity of primers and probes.

### 2.5. Establishment of dddPCR Assay

The dddPCR mixture composition is listed in Table 3, and the operation protocol used was as follows: After all of the solutions were fully mixed, 14 μL of admixture was sucked into the sample port of the chip, which formed a water-in-oil reaction system. The instrument introduced the reaction mixture and mineral oil into the microfluidic chip by the negative pressure method, and then an absolute quantitative analysis was performed by PCR amplification. The thermocycling protocol for the quantification included a 10 min hot start at 95 °C and 40 cycles of PCR (96 °C for 20 s and 60 °C for 60 s). The whole step was completed in a closed environment in the machine, and the test results were obtained by Poisson distribution calculation.

### 2.6. Establishment of Standard Curve

With the purpose of evaluating the reliability of the dual reaction system on the chip, the ddPCR method was used to generate standard curves for the detection of *P. fragi* and *P. aeruginosa*. The linear relationship between the detection of *P. fragi* and *P. aeruginosa* by digital PCR was calculated by adding template concentration 2, 4, 8 and 16 times. The copy value of the sample detection is obtained using the following calculation formula:$$C\left(\mathrm{copies}/\mu L\right)=\frac{P\times V}{V1\times D}$$
where P is the mean software output value, V is the total reaction volume, V1 is the amount of nucleic acid added and D is the dilution multiple.

### 2.7. Sensitivity Test of dddPCR Detection

Genomic DNA sensitivity and bacterial suspension sensitivity of the ddPCR method were tested. The whole genome DNA template of *P. aeruginosa* and *P. fragi* strains were extracted and determined, and the concentrations were serially diluted from 10^6^ to 10^1^ fg/μL. *P. aeruginosa* and *P. fragi* cells were continuously diluted to the final concentration of 10^0^–10^5^ cfu/mL after plant counting. The initial gene concentrations of *P. fragi* and *P. aeruginosa* were 54 ng/μL and 360 ng/μL, respectively, and the numbers of colonies were 110 cfu/mL and 110 cfu/mL. These DNA templates were used for subsequent sensitivity evaluations.

### 2.8. Anti-Interference Ability Evaluation

Bacteria usually coexist in a mixed population in food and environmental samples. In order to evaluate the accuracy of the reaction system in the presence of other interfering bacteria, different concentrations of *P. fragi* and *P. aeruginosa* were mixed with the natural background flora in the collected food samples. To obtain the natural background flora of milk, 25 mL of untreated fresh milk collected from pastures was cultured in 225 mL of LB at 37 °C for 18 h, and the natural background flora of cold fresh chicken was also enriched using this method. Plate counts were performed on all selected bacteria to determine the concentration of cells in the mix and gradually diluted to a concentration of N × 10^2^–10^7^ cfu/mL 1 < N < 10. The counting results showed that the concentration of natural background bacteria in raw milk was 5.4 × 10^7^ cfu/mL, and the concentration of natural background bacteria in chicken was 1.72 × 10^8^ cfu/mL. The genome of the mixed bacteria extracted from the gradient diluted flora was used for the template of dddPCR reaction.

### 2.9. Evaluation of Artificial Simulated Contamination of Actual Samples

To evaluate the applicability of the proposed methods, several foods contaminated by *P. aeruginosa* and *P. fragi* were selected as samples for simulation analysis. *P. aeruginosa* and *P. fragi* were inoculated in drinking water, sterile milk and cold fresh chicken, respectively (initial concentration of inoculation: 10^2^ cfu/mL; inoculation proportion: 10%). The genomes were extracted for dddPCR detection after 0, 3, 6, 9 and 12 h of culture, respectively. All samples underwent the traditional method of microbial culture to ensure that there was no target gene to be detected. The reaction system and conditions were chosen according to the above instructions, and the presence of contamination was evaluated using sterile double distilled water without template control (NTC) reaction.

## 3. Results and Discussions

### 3.1. Analysis of Candidate Gene Selection

A total of 13,613 genes were screened according to the whole genome sequence of three groups of *P. fragi* uploaded from the database, and 38 specific genes were obtained. The homology and coverage of 38 genes of *P. fragi* were 100%, and the homology with other bacteria was very weak. Furthermore, four highly specific regions were found, which could be used as selectable targets. The designed primers were predicted to have no cross-reactivity with other species using the Primer-BLAST tool (https://www.ncbi.nlm.nih.gov/tools/primer-blast/, accessed on 6 July 2023). The primers were designed according to the specific gene, and the specificity of the primers was verified by PCR as a target gene to determine whether the gene could be used as a quasi-specific gene for a subsequent dddPCR reaction. The specificity results, as shown in Table 4, reveal that the *RS22680* gene of *P. fragi* and the *lasR* gene of *P. aeruginosa* had high accuracy and could be used for subsequent dddPCR experiments.

### 3.2. Results of Evaluation of dddPCR Reaction System Construction 

In this experiment, two luminescence channels, FAM and VIC, were selected for the dddPCR assay. The experimental results are shown in Figure 1. The effective droplet generation was greater than 20,000, the negative droplet and the positive droplet were evenly distributed, and the counting was effective. There was no interference in the absolute quantitative detection between the two, and the constructed dual reaction system had excellent detection results. Zhang et al. successfully detected *Salmonella* and Shigella using ddPCR [35]. Luo et al. detected *Staphylococcus aureus* in the mixture using digital PCR [36]. More researchers have focused on the improvement of digital PCR technology. Yin et al. established a multiplex digital PCR method without extracting ctDNA to reduce the reaction steps [37]. Xie Tengbao et al. avoided the interference of cross primers and the overlap of fluorescence in a single tube through physical separation [38]. Simpler and more practical multiple detection methods still urgently need to be developed. The dual channel designed in this assay could accurately detect both microorganisms at the same time, and the luminescent groups used had no interference with each other.

### 3.3. A Linear Relationship Analysis of the Reaction System

The linear relationship of the established dual detection system shows excellent superiority, as seen in Figure 2. A linear correlation of *P. aeruginosa* between the detected and theoretical ratios was obtained with an R^2^ value of 0.9989. Additionally, the obtained and expected values of *P. fragi* showed a good linear correlation (R^2^ = 0.9995). The standard curve for detecting *P. fragi* did not cross the origin. The reason for this phenomenon can be explained as follows: different types of fluorescein dyes have different signal intensities when they are accepted during luminescence and quenching, or there is an advance or delay in the machine acceptance signal [39]. Fluorescein amidites (FAMs), Cyanine (Cy) and carboxy-X-rhodamine (ROX) are the most common fluoresceins, which are received by signals by releasing reporters to change the emission wavelength [40]. By utilizing these changes, the luminescence sensors “off–on” and “on–off” could be used to measure the concentration of the target analyte. Liu et al. designed a dual-channel sensor, with each channel marked distinctly by FAM and ROX [41]. The FAM fluorescein exhibits particularly excellent sensitivity in detection. However, this also leads to background signal interference in the strong fluorescence signal. In terms of accuracy, it is necessary to increase the quality control point as the basis for judging the positive test results. After multiple template-free parallel experiments, the experimental results of the FAM channel were less than 30 copies, and the results were judged to be negative. In contrast, the VIC channel could achieve absolutely accurate detection. When there is no template, the luminescence signal cannot be detected. Bolzon et al. distinguished *Listeria* spp. and *Listeria monocytogenes* by using VIC as the internal control of reaction [42]. Therefore, in the development of multiple detection methods, the selection of fluorescein is also very important. Different types of fluorescein need high accuracy and no interference with each other. In the future, multiple high-sensitivity detection will surely have better development in food safety and public health [43].

### 3.4. Analysis of Specific Primers for dddPCR

In the previous target gene screening experiment, we knew that the *RS22680* gene of *P. fragi* and the *lasR* gene of *P. aeruginosa* had good specificity. The primer and probe of the dddPCR detection system was designed using these two genes, and the results are shown in Figure 3. *P. fragi* and *P. aeruginosa* could be well detected, and other bacteria had no positive reaction. The experimental results show that the primer Pa-2, designed according to the *lasR* gene, had significantly excellent specificity for the detection of *P. aeruginosa.* No cross-reactivity was observed with *Escherichia coli*, *Listeria monocytogenes* and other *Pseudomonas*. In order to detect the accuracy, researchers have developed a lot of methods. *P. aeruginosa* was determined by combining designed targeted crRNA with CRISPR technology [9]. Xiang et al. used cross priming amplification to detect *P. aeruginosa* to determine the accuracy of the results [44]. A fluorescent biosensor combined with the DNAzyme and a new approach using pseudopaline-based probes were designed for the effective detection of *P. aeruginosa* [45,46]. Researchers have always been committed to identifying hazards more accurately through various methods.

### 3.5. Sensitivity Analysis of Genome and Colony Detection by dddPCR

The accuracy of this established dddPCR platform for the simultaneous detection of *P. aeruginosa* and *P. fragi* was evaluated by comparing the measured concentration for each genomic DNA and colony. For the sensitivity evaluation of genomic DNA, the result showed that a weak signal value was generated when the DNA template concentration was lower than 5.4 × 10^3^ fg/μL. Through the previous determination of the control point of FAM fluorescein detection, more than 30 copy values were judged to be positive. Therefore, the lowest detection limit of *P. fragi* was 5.4 × 10^3^ fg/μL. The minimum detection limit of *P. aeruginosa* was 3.6 × 10^2^ fg/μL (Figure 4).

In the sensitivity evaluation of the bacterial suspension template, the detection limits of *P. fragi* and *P. aeruginosa* were all presented in single digits (Figure 5). At present, CRISPR technology is used to detect *P. aeruginosa* with a minimum of 50 cfu/mL [6]. Wang et al. detected *Salmonella* using digital PCR with a sensitivity of 10^−4^ ng/μL or 10^2^ cfu/mL, which are lower values than the results detected in this assay [47]. According to the specific genes screened, the primers designed in this experiment showed a particularly good sensitivity. However, the high sensitivity of digital PCR limits the detection range to a certain extent. When the number of bacterial colonies exceeded 10^4^ cfu/mL, the statistical results were invalid due to sufficient luminescent points when calculating the positive results. By combining a method with a DNAzyme sensor, Qin et al. found that *P. aeruginosa* can reach a concentration of 1.2 cfu/mL [45]. The detection results are comparable to the results of *P. aeruginosa* detection in this paper, and the detection range is wider. However, dddPCR can more easily achieve multiplex detection and a lower cost.

### 3.6. Analysis of Anti-Interference Ability

The purpose of this test was to validate the accuracy of simultaneous detection for *P. fragi* and *P. aeruginosa* under the background microbiota. In this experiment, the natural background flora of fresh milk and chilled meat were selected as the interference factor to explore the accuracy of this method in detecting *P. aeruginosa* and *P. fragi* in the case of rich microbial species. The bacterial concentration of *P. aeruginosa* was 1.5 × 10^8^ cfu/mL after culture, and that of *P. fragi* was 1.8 × 10^8^ cfu/mL. After ten-fold gradient dilution, it was used for an anti-interference experimental analysis. The results show that different natural background flora have no effect on the detection of *P. aeruginosa*. As Figure 6 shows, the sensitivity of *P. fragi* was slightly affected under the natural background flora concentration of milk at 10^3^ cfu/mL. This result might be due to the existence of strains in fresh milk that have a greater growth advantage than *P. fragi*, and there was a phenomenon of competitive inhibition. Previous studies have shown that *E.coli* was able to coexist with spoilage *Pseudomonas*, which could lead to meat food spoilage and has a clear leading role compared to other microorganisms [48,49]. We guess that microorganisms in fresh milk may have more dominant strains, which may affect the detection of *P. fragi*. Whether the concentration of the background flora will affect the sensitivity of the detection was also verified. When the concentration of natural background bacteria was higher or lower than 10^3^ cfu/mL, the detection results were stable. This indicates that the detection method we established still has great sensitivity even under the interference of other background bacteria.

### 3.7. Analysis of Test Results of Artificially Contaminated Food

To evaluate the feasibility and reliability of the dddPCR assay, the detection of *P. fragi* and *P. aeruginosa* in artificially spiked samples was performed. *P. aeruginosa* and *P. fragi* obtained from the overnight culture were inoculated in nutrient broth at a ratio of 10% to achieve artificially contaminated samples with an initial contamination level of N × 10^0^. The results show that the target bacteria could not be detected without an enrichment culture (Figure 7). After 9 h of culture in milk, the growth of *P. aeruginosa* and *P. fragi* was significantly higher than that in chilled chicken or drinking water. This result suggests that nutritious and uniform food is more conducive to the growth of microorganisms, and this type of food should be regularly controlled and monitored. The results show that cold fresh chicken is more susceptible to microbial infection between 0 and 6 h after inoculation with *Pseudomonas*, which also proves that the water activity of raw meat is more likely to nourish bacteria. A longer detection time may reflect the growth status of Pseudomonas in different substrates, but the high sensitivity of ddPCR limits the wide range of detection. The experimental results also show that there were great differences in the growth status of the two bacteria, even if both bacteria belonged to the *Pseudomonas* genus. *P. fragi* has the advantage of long-term survival in the environment. On the other hand, it is easier to achieve the early control of *P. aeruginosa*.

Although recent detection methods have made great progress, there is still a lot of room for development in the detection and analysis of hazardous substances in food. The composition of food will still greatly affect the accuracy and sensitivity of detection. Future research will still focus on stability, sample pretreatment and mutual interference. The development of multiple detection methods that can meet the current detection needs is of great significance for food safety monitoring.

## 4. Conclusions

Based on two-color fluorescent probes, we developed a dddPCR detection system that detected *P. aeruginosa* and *P. fragi* simultaneously. Both bacteria were accurately detected using the dddPCR method, and the mean R^2^ values were greater than 0.95. Furthermore, the sensitivity was higher than that of other reported PCR techniques. No significant effect on the test results was observed in the presence of natural background bacteria in chicken or milk. Moreover, this method can detect 10^0^ cfu/mL of bacterial DNA, and the detection genomics DNA limit was 10^2^ fg/μL. The applicability of artificially infected drinking water, milk and chicken samples was evaluated. Both pathogens were successfully detected after 3 h of contamination. This method has great potential for detecting food safety and ensuring product quality. It is necessary to encourage factories to use this method for product supervision. However, as this method has only been tested under laboratory conditions, there is still much room to optimize the detection of multiple pathogens on the market. In the future, a digital PCR reaction system could be established for multiple pathogenic bacteria detection based on different products in order to achieve the rapid and accurate detection of actual products on site.

## Figures and Tables

**Figure 1 foods-13-01453-f001:**
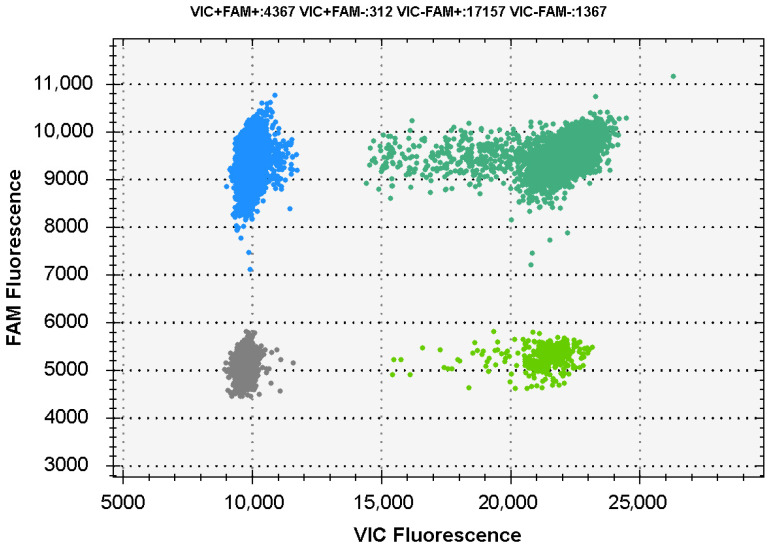
The results of the establishment of a dual detection system. Note: 
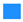
 represents positive *P. fragi* reaction, 
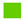
 represents positive *P. aeruginosa* reaction, 
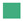
 represents positive *P. fragi* and *P. aeruginosa* reaction and 
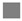
 represents negative *P. fragi* and *P. aeruginosa* reaction.

**Figure 2 foods-13-01453-f002:**
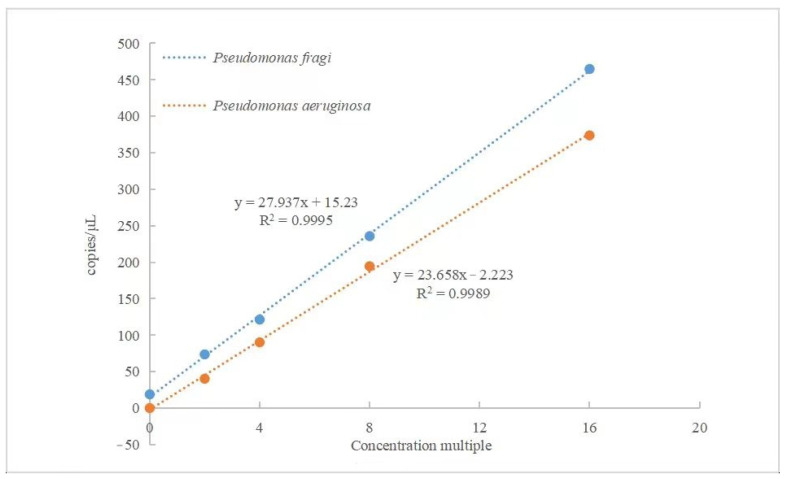
Linear relationship analysis of *P. fragi* and *P. aeruginosa* detected using dddPCR method.

**Figure 3 foods-13-01453-f003:**
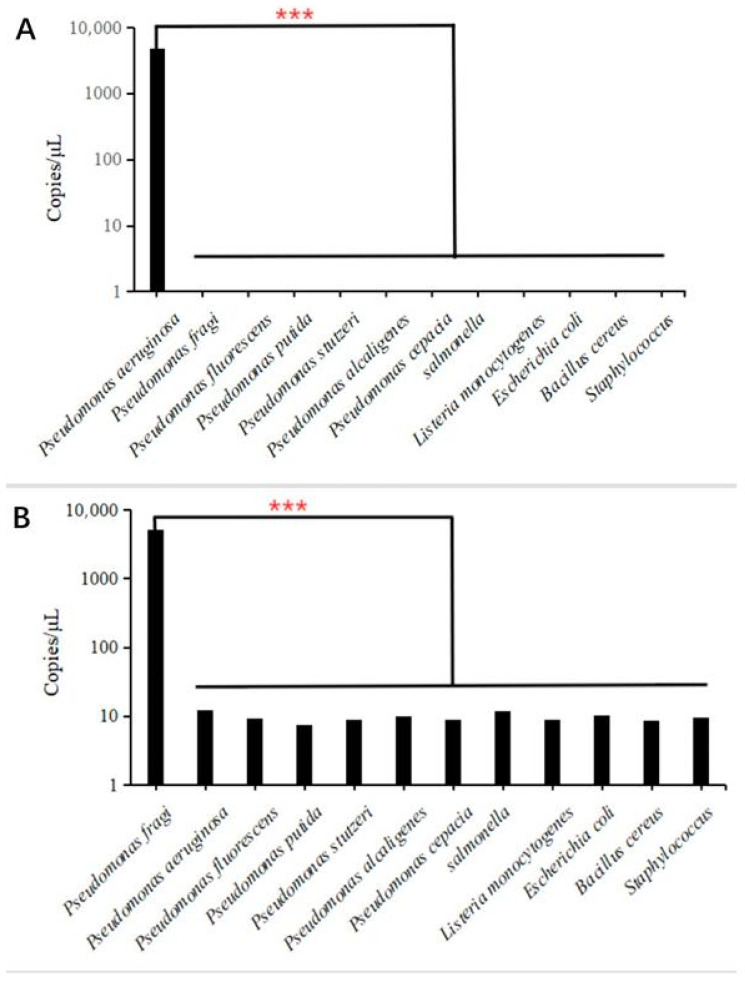
Specific results of dddPCR detection of *P. aeruginosa* and *P. fragi.* Note: (**A**) specific results of *P. aeruginosa*; (**B**) specific results of *P. fragi.* ***: The result is significant.

**Figure 4 foods-13-01453-f004:**
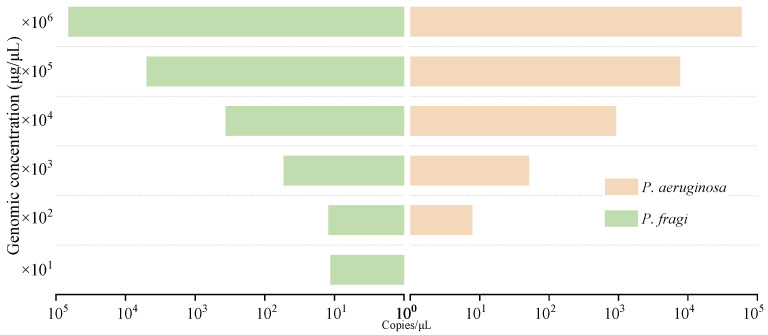
Genomic sensitivity analysis of dddPCR assay.

**Figure 5 foods-13-01453-f005:**
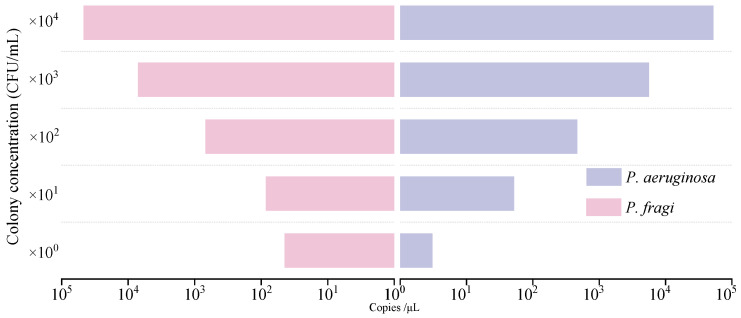
Colonial sensitivity analysis of dddPCR assay.

**Figure 6 foods-13-01453-f006:**
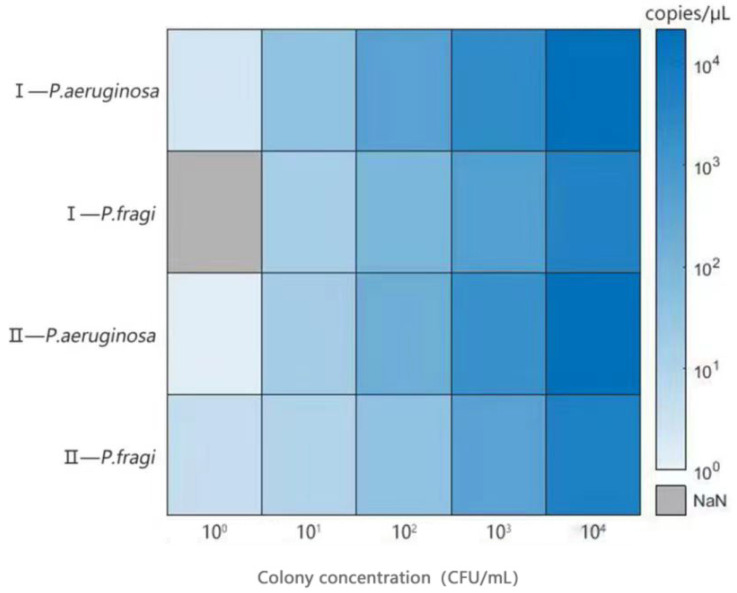
Sensitivity evaluation of ddPCR method in presence of natural background flora in food. Note: Ⅰ: concentration of natural background bacteria in milk was 5.4 × 10^3^ cfu/mL; II: concentration of natural background bacteria in chicken was 1.72 × 10^3^ cfu/mL; NaN: invalid result.

**Figure 7 foods-13-01453-f007:**
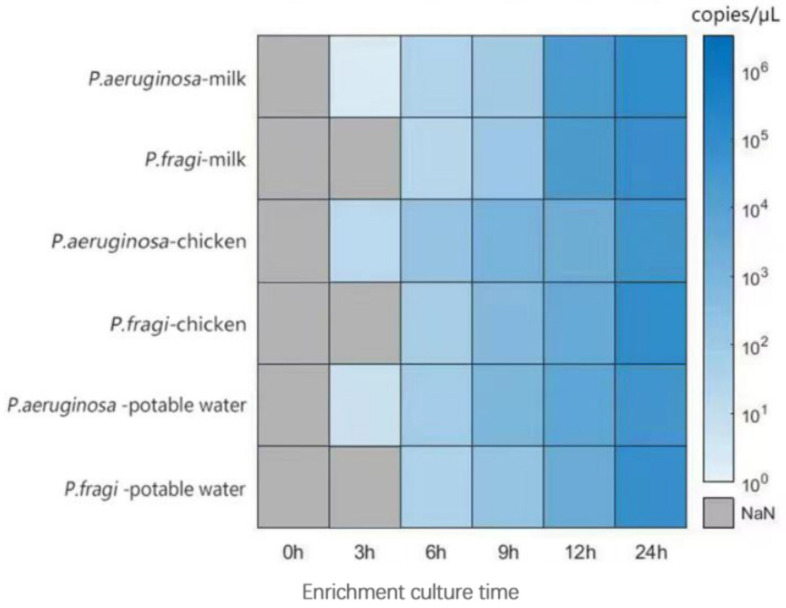
Analysis of detection results of artificial *P. fragi* and *P. aeruginosa* in food.

**Table 1 foods-13-01453-t001:** The candidate genes and primers of *P. aeruginosa* and *P. fragi* that were used in Pthe CR specificity verification experiments.

Source	Gene	Annotation	Primer	Sequence (5′–3′)	Source
*P. fragi*	*RS22665*	Transcriptional regulatory protein	pf1-7	ATAACGGCAAGAACACCA	In this study
				CCAAACACGCCTCTGAAC	
	*RS22680*	NeuD/PglB/VioB family sugar acetyltransferase	pf1-18	GGCACAAGTCAATGGTCG	
				CACAGTCAGGGCAAGGAT	
	*RS10890*	triacylglycerol lipase	Pf3-21	CCTTGAATGCGCTTAACGCCCTGACCACC	
				CGTAGACCCGGTCCAGTAGGCGAGGCTGAT	
	*ribA*	GTP cyclohydrolase II	Pf3-13	CGATGTATTCGGGTCCAGACGCTGTGATT	
				ATAGTGGTAGTTGTCTTGGGACGGTAGGC	
*P. aeruginosa*	*lasR*	Transcriptional regulatory protein	lasR1	CGAGAACGCCTTCATCGTCGGCAACTACC	In this study
				GAAGAACTCGTGCTGCTTTCGCGTCTGGTA	
	*gyrB*	DNA gyrase subunit B	gyrB2	ATCCGCACCCTGCTGTTGACCTTCTTCTTCCG	
				TGATGTACTGCTCCTGCTTGCCACGCTTGACC	
	*rpoB*	DNA-directed RNA polymerase beta chain	rpoB3	TGCCCGATCGAAACCCCTGAAGGTCCGAA	
				ATCTCGTCGGTTACCAGGCTGTCCTTGACT	

**Table 2 foods-13-01453-t002:** The primers and probes in the dddPCR assay for the detection of *P. aeruginosa* and *P. fragi*.

Bacterial Strains	Gene	Primer	Sequences of Primer (5′–3′)	PCR Product	Source
*P. aeruginosa*	*lasR*	Pa-2F	AGCCGGGAGAAGGAAGTGTT	80 bp	In this study
		Pa-2R	TCCGAGCAGTTGCAGATAACC		
		Pa-2P	VIC-TGCGCCATCGGCAAGACCAGT-BHQ1		
*P. fragi*	*RS22680*	Pf-2F	GGCCGGCACGCAAGT	59 bp	In this study
		Pf-2R	CTTGGACAGTAGCGAAAAACGA		
		Pf-2P	FAM-TGTCGAGAAGCCAGTCTCCGTGTTCC-BHQ1		

**Table 3 foods-13-01453-t003:** The reaction system of the dddPCR assay.

Component	Addition
5× MIX	4.5 μL
Primer 1-F (10 μM)	1 μM
Primer 1-R (10 μM)	1 μM
Primer 2-F(10 μM)	1 μM
Primer 2-R (10 μM)	1 μM
Probe1 (10 μM)	0.25 μM
Probe2 (10 μM)	0.25 μM
ROX dye	0.3 μL
Enzyme	0.2 μL
Template 1	1 μL
Template 2	1 μL
Complemented by water to	15 μL

**Table 4 foods-13-01453-t004:** The results of the PCR analysis of species-specific genes.

Bacterial Strains	Source	Results						
		*lasR*	*rpoB*	*gyrB*	*RS22665*	*RS22680*	*RS10890*	*ribA*
*Pseudomonas fragi*	SHBCC D24613	−	−	−	+	+	+	+
*Pseudomonas fragi*	CGMCC1.3349	−	−	−	+	+	+	+
*Pseudomonas fragi*	Laboratory isolates	−	−	−	+	+	+	−
*Pseudomonas aeruginosa*	ATCC 15442	+	+	+	−	−	−	−
*Pseudomonas aeruginosa*	ATCC 27853	+	+	+	−	−	−	−
*Pseudomonas aeruginosa*	DSM 939	+	+	+	−	−	−	−
*Pseudomonas aeruginosa*	Laboratory isolates	+	+	+	−	−	−	−
*Pseudomonas fluorescens*	ATCC 13525	−	−	−	+	−	+	−
*Pseudomonas putida*	ATCC 49128	−	+	−	−	−	−	+
*Pseudomonas pseudoalaligenes*	CGMCC1.10611	−	−	−	−	−	−	−
*Pseudomonas mendocina*	ATCC 25411	−	+	−	−	−	−	+
*Pseudomonas stutzeri*	ATCC 17588	−	−	−	−	−	−	−
*Pseudomonas alcaligenes*	ATCC 14909	−	−	−	−	−	−	−
*Pseudomonas cepacia*	SHBCC D 14769	−	−	−	−	−	−	−
*Pseudomonas putida*	ATCC 17485	−	−	−	−	−	−	−
*Pseudomonas fluorescens*	GIM1.110	−	−	−	−	−	+	−
*Pseudomonas fluorescens*	ATCC 17397	−	−	+	−	−	−	−
*Staphylococcus*	CICC 10788	−	−	−	−	−	−	−
*Enterococcus avium*	ATCC 14025	−	−	−	−	−	−	−
*Bacillus pumilus*	CMCC 63202	−	−	−	−	−	−	−
*Listeria monocytogenes*	CICC 21622	−	−	−	−	−	−	−
*salmonella enterica*	CICC 21482	−	−	−	−	−	−	−
*Cronobacter sakazakii*	CICC 21560	−	−	−	−	−	−	−
*Cronobacter universalis*	NCTC 9529	−	−	−	−	−	−	−
*salmonella anatum*	CICC 21498	−	−	−	−	−	−	−
*Escherichia coli*	ATCC 25922	−	−	−	−	−	−	+
*Bacillus cereus*	CICC 23384	−	−	−	−	−	−	−

Note: +: positive result; −: negative result.

## Data Availability

The original contributions presented in the study are included in the article, further inquiries can be directed to the corresponding author.

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
