# Peer review of "An Evaluation of the Sensitivity and Applicability of a Droplet Digital Polymerase Chain Reaction Assay to Simultaneously Detect Pseudomonas aeruginosa and Pseudomonas fragi in Foods"

_foods, 2024, doi:10.3390/foods13101453_

Round 1
Reviewer 1 Report
Comments and Suggestions for Authors
Few points to be addressed
- It is necessary to clearly explain why the study focuses on the genus Pseudomonas. The article quickly transitions to discussing diagnostic practices, scattering mentions about the relevance of Pseudomonas as a pathogen. I suggest adding a paragraph in the introduction addressing this topic and consolidating this information there.
- The methodology is well-organized and described satisfactorily.
- Since the article concerns the ddPCR technique for detecting Pseudomonas, it would be interesting to discuss more broadly the sensitivity and specificity compared to the conventional method (microbiological), as well as to present the pros and cons of both techniques.
- In Table 4, some Pseudomonas species did not present any of the investigated genes. How, then, is the diagnosis made for these species? It would be interesting to present an alternative or even other genes of interest if they are species with potential pathogenicity.
- In Figure 7, page 11, line 354, the species names are not in italics.
Author Response
|
Response to Reviewer 1 Comments
|
||
|
1. Summary |
|
|
|
Thank you very much for taking the time to review this manuscript. Please find the detailed responses below and the corresponding revisions/corrections highlighted/in track changes in the re-submitted files. |
||
|
2. Questions for General Evaluation |
Reviewer’s Evaluation |
Response and Revisions |
|
Does the introduction provide sufficient background and include all relevant references? |
Can be improved |
|
|
Are all the cited references relevant to the research? |
Can be improved |
|
|
Is the research design appropriate? |
Can be improved |
|
|
Are the methods adequately described? |
Can be improved |
|
|
Are the results clearly presented? |
Can be improved |
|
|
Are the conclusions supported by the results? |
Can be improved |
|
|
3. Point-by-point response to Comments and Suggestions for Authors |
||
|
Comments 1: It is necessary to clearly explain why the study focuses on the genus Pseudomonas. The article quickly transitions to discussing diagnostic practices, scattering mentions about the relevance of Pseudomonas as a pathogen. I suggest adding a paragraph in the introduction addressing this topic and consolidating this information there. |
||
|
Response 1: Thank you for your suggestions on the content of the paper. The introduction of Pseudomonas and its diagnostic methods are improved in the introduction. This makes the article more closely linked and the expression more reasonable. This part can be found – page 1, paragraph 1. |
||
|
Comments 2: The methodology is well-organized and described satisfactorily |
||
|
Response 2: Thank you very much for your approval of the experimental method designed in this paper! We will continue to work hard in the future. |
||
|
Comments 3: Since the article concerns the ddPCR technique for detecting Pseudomonas, it would be interesting to discuss more broadly the sensitivity and specificity compared to the conventional method (microbiological), as well as to present the pros and cons of both techniques |
||
|
Response 3: We kindly agree with your idea that the traditional microbial culture detection method and the ddPCR detection method are very different in specificity and sensitivity. However, there are too many contents to discuss the advantages and disadvantages between the two, such as the isolation and identification of different bacteria, the identification of different serotypes or different species of the same bacteria. The detailed discussion of this part may lead to problems such as logical confusion, language verbosity, and excessive length of the preface. In this paper, the current situation of traditional detection methods is briefly introduced, and the application of ddPCR technology is also explained. Can we discuss this aspect with two detection methods of a Pseudomonas ? For example, the traditional microbial culture detection of Pseudomonas aeruginosa is based on its specific growth, morphology and physiological and biochemical characteristics on a specific medium. The suspected colonies were confirmed by colony color, fluorescence characteristics, acetamide broth, oxidase and other biochemical tests. This is extremely easy to cause false negative results, and the specificity is not high. For the emergence of non-Pseudomonas aeruginosa bacteria also need to pass other complex biochemical identification. |
||
|
Comments 4: In Table 4, some Pseudomonas species did not present any of the investigated genes. How, then, is the diagnosis made for these species? It would be interesting to present an alternative or even other genes of interest if they are species with potential pathogenicity. |
||
|
Response 4: Thank you very much for your suggestion. Molecular testing is still possible for other investigated genes mentioned in Table 4 that have not been reported in Pseudomonas. The whole genome of this species can be obtained by sequencing. According to the method in this paper, diagnosis can be completed by comparing genome sequence, screening specific genes and designing primers. If it is a potentially pathogenic species, the functional identification analysis of its genes is indeed of high value. We will continue to take a closer look on this basis. Thanks again for your advice! |
||
|
Comments 5: In Figure 7, page 11, line 354, the species names are not in italics. |
||
|
Response 5: Thank you very much for pointing out the problem, we have changed the mentioned species names in italics. This correction can be highlighted in the text. |
||
Reviewer 2 Report
Comments and Suggestions for Authors
Comments and notes can be seen in the text.

Author Response
|
Response to Reviewer 2 Comments |
||
|
1. Summary |
|
|
|
Thank you very much for taking the time to review this manuscript. For your suggestions and questions, please find the detailed responses below and the corresponding revisions/corrections highlighted/in track changes in the re-submitted files. |
||
|
2. Questions for General Evaluation |
Reviewer’s Evaluation |
Response and Revisions |
|
Does the introduction provide sufficient background and include all relevant references? |
Can be improved |
|
|
Are all the cited references relevant to the research? |
Yes |
|
|
Is the research design appropriate? |
Yes |
|
|
Are the methods adequately described? |
Can be improved |
|
|
Are the results clearly presented? |
Must be improved |
|
|
Are the conclusions supported by the results? |
Yes |
|
|
3. Point-by-point response to Comments and Suggestions for Authors |
||
|
Comments 1: Must be in italics |
||
|
Response 1: Thank you for pointing this out. We agree with this comment. Therefore, we have put all the species mentioned in italics. |
||
|
Comments 2: A space is missing |
||
|
Response 2: Thank you very much for your careful review of this paper. We have rechecked and corrected the missing or redundant spaces in the text. |
||
|
Comments 3:The abstract must follow the journal’s requirements:background and aim are unclear.Must improve it |
||
|
Response 3: Thank you for pointing this out. We give a more specific explanation of the background and purpose of the abstract. -Page 1, Line 10 |
||
|
Comments 4: You must use the SI for the units |
||
|
Response 4: Thank you for pointing out the problem, we replaced CFU/mL with cfu/mL. |
||
|
Comments 5: ddPCR:Which one is the correct?dddPCR or ddPCR? |
||
|
Response 5: Thank you very much for pointing out the problem, which is indeed a mistake left in the process of writing the paper. The full text has been revised to dddPCR. |
||
|
Comments 6: Format |
||
|
Response 6: Thank you for the format of the reference footnotes. We have revised the same questions in the full text. |
||
|
Comments 7: Improve redaction |
||
|
Response 7: Thank you for your suggestions for this sentence, we have modified it. The sentence is modified as : In addition, due to the rich biofilm, P. fragi can easily contaminate aquatic products such as salmon.-Page 2, Line 62 |
||
|
Comments 8: Low temperature must tell which temperature |
||
|
Response 8: Thank you for the question. According to the existing literature, Pseudomonas fragi can grow as low as 4 °C. However, the strains isolated from glaciers described in this paper have no detailed information. For the sake of rigorous consideration of the presentation of the paper, we decided to delete the sentence that could not prove in detail how low the temperature at which Pseudomonas fragi could grow. |
||
|
Comments 9: Improve redaction What’s the meaning of less |
||
|
Response 9: Thank you for pointing out the problem, we also believe that the number of statements is not rigorous. Changes have been made, which can be seen in the highlighted text. |
||
|
Comments 10: Must put place and the coordinates of the site |
||
|
Response 10: Thank you very much for your suggestions. We added the address information in more detail. The specific performance is : Samples (raw chicken and potable water) were purchased from a fresh supermarket in Fuxi Street, Fengyang County, Chuzhou City, Anhui Province, China. -Page 3, Line 97 |
||
|
Comments 11: Add coordinates |
||
|
Response 11: Thank you very much for your suggestions. We added specific address information. The original text is modified as follows : Fresh milk was obtained from Dutch dairy cows (Heping Dairy Ranch, No.1151, South Yan 'an Road, Bengbu City, Anhui Province, China) through aseptic sampling and sent back to the laboratory for processing as soon as possible at low temperature. -Page 3, Line 99 |
||
|
Comments 12: Small pieces :specify size |
||
|
Response 12: Thank you for your questions. For the problem of inaccurate expression here, we have made changes. In the experiment, we divided the chicken into 25 g of samples according to the actual situation, and there was no fixed size. |
||
|
Comments 13: Culture method: Add reference of the cultivation method used |
||
|
Response 13: |
||
|
Comments 14: Names of genes in lowercase |
||
|
Response 14: Thank you for pointing out this professional problem. We have replaced LasR with lasR.-Page 4, Line 149 |
||
|
Comments 15: Table :Change the color and delete underline in the table |
||
|
Response 15: Thank you for your careful reading of this manuscript. We modified it according to the problem you pointed out. |
||
|
Comments 16: Change the unit format |
||
|
Response 16: Thank you for your question. The problem of unit format has been modified in full text, which can be seen in the highlighted annotation of the new text. |
||
|
Comments 17: Rectify the format |
||
|
Response 17: Thank you for pointing out the formatting problem, which has been corrected. -Page |
||
|
Comments 18: Specifies which signal corresponds to each detection |
||
|
Response 18: Thank you very much for pointing out the problem ! The description has been added to the notes below the chart. -Page 6, Line 213 |
||
|
Comments 19: Add refernces |
||
|
Response 19: Thank you for your suggestions. This sentence is a representation of the results of the linear relationship in Figure 2, and there is no reference. The references of relevant discussions are marked. |
||
|
Comments 20: Add more refernces or improve redaction |
||
|
Response 20: Thank you for your suggestions for this part. We have made a more rigorous expression of the sentence, which can be seen in the newly uploaded text.-Page 7, Line 225 |
||
|
Comments 21: Must reference only the last name |
||
|
Response 21: Thank you for pointing out this problem. We are very sorry for our carelessness and have made corresponding modifications. -Page 8, Line 260 |
||
|
Comments 22: Figure2: Improve quality of this figure and adjust the range of the axis |
||
|
Response 22: Thank you for your question. We readjusted the coordinates according to your suggestion, which makes the picture clearer. |
||
|
Comments 23: Must improve quality and a bigger size of the figure |
||
|
Response 23: Thank you for your question. The picture size is really small. We have made adjustments, which can be found in the new text. |
||
|
Comments 24: The detection limit in x10^3 does not coincide with that represented in Figure 4. Must improve |
||
|
Response 24: Thank you very much for your question. The detection limits in the figure show only orders of magnitude. The specific colony numbers of P. aeruginosa and P. fragi are indicated in method 2.7. -Page 5, Line 177 |
||
|
Comments 25: Must use the same format. With or without lines. Delete the line from the correction |
||
|
Response 25: Thank you very much for pointing out the problem, the revision line that appeared in the article has been removed. |
||
|
Comments 26: Figure6: For better orientation it is suggested to place the X axis |
||
|
Response 26: Thank you very much for pointing this out, we take your suggestion to add the X-axis explanation |
||
|
Comments 27: Figure7:This format of the Y axis is more understandable than Figure 6. As a suggestion in Figure 6, add the sample where the microorganism in question comes from |
||
|
Response 27: Thank you for your suggestion. We also considered placing the Y-axis in this way before, but the picture was not beautiful due to too many annotations, so the explanation was put in the comment under the picture. |
||
|
Comments 28: Must follow the format for references from the journal 1. Author 1, A.B.; Author 2, C.D. Title of the article. Abbreviated Journal Name Year, Volume, page range. |
||
|
Response 28: Thank you very much for your questions. We have reviewed and revised the format of the reference twice. |
||
Reviewer 3 Report
Comments and Suggestions for Authors
The article entitled "Evaluation of sensitivity and applicability of a Droplet Digital PCR assay for simultaneous detecting Pseudomonas aeruginosa and Pseudomonas fragi in foods" has 14 pages, 45 References, 7 figures and 4 Tables.
The manuscript of the article is written in a generally declared manner and meets the requirements for printing.
From a formal point of view, there are minor typos in the manuscript of the article, but these will be removed as part of the proof-of-reading. These are, for example, missing or redundant spaces in parentheses,
In terms of content, I have no comments, apart from one fact that is repeated in a large number of articles:
"Samples(raw chicken and potable water) were purchased in a supermarket in Chu-Zhou, Anhui, China. Fresh milk was obtained from Dutch dairy cows ( Heping Dairy Ranch, Bengbu City, Anhui Province, China ) through aseptic sampling and sent back to the laboratory for processing as soon as possible at low temperature. The raw chicken was cut into small pieces and frozen at -20℃ for subsequent experiments. For the latter experiments of artificial pollution commercial sample, purchased drinking water, sterile milk and raw chicken were identified by microbial culture method without P. aeruginosa or P. fragi."
The type and origin of the material, its preparation for the experiment is described so briefly that it is impossible to repeat the experiment or say what specifically the scientists from the author's team analyzed.
Author Response
|
Response to Reviewer 3 Comments |
||
|
1. Summary |
|
|
|
Thank you very much for taking the time to review this manuscript. Please find the detailed responses about the method description of sample below and the corrections highlighted. Thank you for your rigorous attitude to science, and we will deeply reflect on the problems caused by our over simplification. |
||
|
2. Questions for General Evaluation |
Reviewer’s Evaluation |
Response and Revisions |
|
Does the introduction provide sufficient background and include all relevant references? |
Yes |
|
|
Are all the cited references relevant to the research? |
Yes |
|
|
Is the research design appropriate? |
Can be improved |
|
|
Are the methods adequately described? |
Must be improved |
|
|
Are the results clearly presented? |
Can be improved |
|
|
Are the conclusions supported by the results? |
Can be improved |
|
|
3. Point-by-point response to Comments and Suggestions for Authors |
||
|
Comments 1: The type and origin of the material, its preparation for the experiment is described so briefly that it is impossible to repeat the experiment or say what specifically the scientists from the author's team analyzed. |
||
|
Response 1: We sincerely thank you for your comments. We agreed that the description in this section is too simple. We accepted your suggestion and made more detailed changes to the description of this method. More sample processing steps can be seen in the specific experimental methods (2.8 & 2.9). The specific update content is: Samples (raw chicken and potable water) were purchased from a fresh supermarket in Fuxi Street, Fengyang County, Chuzhou City, Anhui Province, China. The raw chicken was cut into 25 g and frozen at -20 ℃ for subsequent experiments. Fresh milk was obtained from Dutch dairy cows (Heping Dairy Ranch, No.1151, South Yan 'an Road, Bengbu City, Anhui Province, China) through aseptic sampling and sent back to the laboratory for processing as soon as possible at low temperature. For the latter experiments of artificial pollution commercial sample, purchased drinking water, sterile milk and raw chicken were identified by microbial culture method (ISO 22717: 2015) without P. aeruginosa or P. fragi. |
||